



# Long-term impacts of global temperature stabilization and overshoot on exploited marine species

Anne L. Morée[1,2], Fabrice Lacroix[1,2,3], William W. L. Cheung[4], Thomas L. Frölicher[1,2]

[1]Climate and Environmental Physics, Physics Institute, University of Bern, Bern, Switzerland
[2]Oeschger Centre for Climate Change Research, University of Bern, Bern, Switzerland
[3]Institute of Geography, University of Bern, Bern, Switzerland
[4]Institute for the Oceans and Fisheries, The University of British Columbia; Vancouver, Canada.

*Correspondence to*: Thomas L. Frölicher(thomas.froelicher@unibe.ch)

**Abstract.** Global warming alters ocean conditions, which can have dramatic consequences for marine species. Yet, the
centennial-scale effects and reversibility of habitat viability for marine species, particularly those that are important to fisheries, remain uncertain. Using the Aerobic Growth Index, we quantify the impacts of warming and deoxygenation on the contemporary habitat volume of 46 exploited marine species in novel temperature stabilization and overshoot simulations until 2500. We demonstrate that only around half the simulated loss of contemporary (1995-2014) habitat volume is realized when warming levels are first reached. Moreover, in an overshoot scenario peaking at 2°C global warming before stabilizing at
1.5°C, the maximum decrease in contemporary habitat volume occurs more than 150 years post-peak warming. Species' adaptation may strongly mitigate impacts depending on adaptation rate and pressure. According to our study, marine species will be affected for centuries after temperature stabilization and overshoot, with impacts surpassing those during the transient warming phase.

## 1 Introduction

Marine fisheries play a pivotal role in the global food system, generating economic benefits (Sumaila et al., 2012), supplying essential nutrients to humans (Hicks et al., 2019) and supporting people's livelihoods (Teh & Sumaila, 2013). Recent international assessments emphasize the predominately adverse impacts of climate change on marine ecosystems and fisheries (Bongaarts, 2019; IPCC, 2019, 2021). Projections indicate that ocean warming (Cheng et al., 2022) and deoxygenation
(Frölicher et al., 2009; Kwiatkowski et al., 2020; Schmidtko et al., 2017) will likely result in a global decline in total animal biomass (Lotze et al., 2019), potential fisheries catch (Cheung et al., 2016) and alterations in species distributions (Deutsch et al., 2020; Hodapp et al., 2023; Morée et al., 2023). Despite considerable research on the transient impacts of global warming and deoxygenation throughout the 21st century on marine fisheries, there remains a considerable gap in understanding the long-term repercussions of centennial-scale warming and deoxygenation. Moreover, the increasing probability of a temporary
temperature overshoot, despite efforts to limit global warming to the Paris Agreement's 1.5°C long-term goal (Rogelj et al.,



2018), raises critical concerns - particularly regarding our limited understanding of the potential reversible or irreversible impacts of such a temporary overshoot on marine species (Meyer et al., 2022; Meyer & Trisos, 2023).

To comprehensively assess the multi-centennial impacts on marine species and potential reversibility after a temporary temperature overshoot, policy-relevant multi-centennial temperature stabilization and overshoot simulations are needed, but

are currently lacking (King et al., 2021; "Reversing climate overshoot," 2023). Recent progress has been made in understanding the physical and biogeochemical reversibility and hysteresis under idealized scenarios (Frölicher & Joos, 2010; Jeltsch-Thömmes et al., 2020; Schwinger et al., 2022). These studies are, however, not extended to species impacts and are not designed to achieve specific global warming levels (Keller et al., 2018). Modelling studies that consider impacts on global marine species are often limited to transient warming (i.e., a non-stabilized and therefore temporary warming and oxygen level

(Cheung et al., 2016; Deutsch, Ferrel, Seible, et al., 2015; IPCC, 2019; Morée et al., 2023)) or idealized overshoot scenarios (Meyer et al., 2022; Santana-Falcón et al., 2023), and do not apply to specific species (Deutsch et al., 2020; Oschlies, 2021; Santana-Falcón et al., 2023). Moreover, modelling of species' distributions generally relies on the environmental state of the sea surface or sea floor only (Garciá Molinos et al., 2016; Heneghan et al., 2021; Hodapp et al., 2023; Meyer et al., 2022), thereby neglecting the crucial vertical dimension when estimating changes in marine species distributions (GA, 2017) and

hence largely neglecting changes in the ocean interior for disturbances in marine species distributions.

Recent advances in climate scenarios and impacts assessments on marine species under ocean warming and deoxygenation, such as the Adaptive Emission Reduction Approach (AERA; Terhaar et al., 2022) and the Aerobic Growth Index (AGI; Clarke et al., 2021; Morée et al., 2023), enable temperature-targeted modelling of the viability of contemporary distributions of marine species. AERA achieves policy-relevant temperature stabilization at any desired level as well as temporary temperature

overshoot at any desired magnitude and duration by iteratively adjusting $CO_2$ forcing equivalent ($CO_2$-fe) emissions in Earth system models. AGI, when above the species-specific critical level ($AGI^{crit}$), signifies the potential for a viable population of the considered species to be sustained with respect to temperature and the partial pressure of oceanic oxygen, $pO_2$. This makes AGI a metric for habitat viability (Clarke et al., 2021; Morée et al., 2023).

Here, we apply AERA to the Earth system model GFDL-ESM2M to simulate novel global warming scenarios spanning the

period 1861 to 2500, encompassing stable temperatures of 1.2℃, 1.5℃, 2℃, and 3℃ as well as 2℃ and 3℃ overshoot scenarios followed by a return to stabilized 1.5℃ warming relative to the 1861-1900 baseline (Fig. 1a,b). We quantify changes in the contemporary habitat volume of 46 representative marine species that are important to fisheries across the epipelagic, mesopelagic, and demersal realms by utilizing AGI. Additionally, we examine the potential effects of species' adaptation to ocean warming and deoxygenation on their habitat volume loss, as adaptation is likely an important factor at the centennial

time scale considered here (Pinsky et al., 2020).






## 2 Methods

### 2.1 Earth System Model

The temperature stabilization and overshoot simulations were carried out with the fully coupled Earth system model GFDL-ESM2M (Dunne et al., 2012, 2013). The GFDL-ESM2M couples an ocean, atmosphere, land, and sea-ice model. Ocean
physics, including sea ice, is simulated with Modular Ocean Model version 4p1, at a 1° horizontal resolution which increases to up to ⅓° meridionally at the equator and is tripolar above 65°N, with 50 vertical levels (Griffies, 2012). Mesoscale eddies are parameterized. Marine biogeochemistry is represented by Tracers of Ocean Productivity with Allometric Zooplankton version 2 (TOPAZv2) and consists of the cycles of carbon, nitrogen, phosphorus, silicon, iron, $O_2$, alkalinity, and lithogenic material as well as surface sediment calcite. Three phytoplankton and one zooplankton group are explicitly represented in
TOPAZv2. Denitrification is included under suboxic conditions, while in the absence of both $O_2$ and nitrate additional respiration is accumulated as a negative $O_2$ concentration. The atmospheric module (AM2) consists of 24 vertical levels and a horizontal resolution of 2° latitude × 2.5° longitude (Anderson et al., 2004). The land model (LM3.0) represents the hydrological, energy and carbon cycles on land.

### 2.2 Model Simulations with the Adaptive Emissions Reduction Approach

The model employed the AERA (Terhaar et al., 2022) to conduct novel simulations spanning from 1861 to 2500 that stabilize global mean 2-m air temperature anomalies at 1.2℃, 1.5℃, 2℃, and 3℃, including temporary overshoots to 2℃ and 3℃ and then returning to stable 1.5℃ warming relative to 1861-1900. All simulations branch off an emissions-driven simulation over the historical period from 1861 to 2005. Post-2005, fossil fuel $CO_2$ emissions follow observed emissions until 2020 and projected emissions from the Nationally Determined Contributions (Climate Action Tracker:
https://climateactiontracker.org/global/temperatures/, last accessed December 2021) from 2021 to 2025. From 2025 onward, prescribed fossil-fuel $CO_2$ emissions are obtained every five years from AERA by subtracting $CO_2$ forcing equivalent ($CO_2$-fe) emissions from prescribed non-$CO_2$ radiative agents and land use change from the AERA-derived total $CO_2$-fe emission curve. Non-$CO_2$ radiative forcing and land use change are prescribed to follow the RCP2.6 scenario till 2100, with no changes thereafter. The AERA approach adapts $CO_2$-fe emissions successively to converge to pre-defined warming levels. AERA
calculates every five years the realized simulated anthropogenic warming and the resulting remaining $CO_2$-fe emission budget. This remaining emission budget is then distributed across future years using a cubic polynomial function. The temporary overshoot simulations are branched off from the 2℃ and 3℃ stabilization simulations, gradually reducing the warming level to 1.5℃ by circa 2300 following the approach described in Terhaar et al. (2022). The resulting $CO_2$-fe emission pathways that



forced the model to the different warming levels show a maximum change in annual $CO_2$-fe emissions of -2.1, -1.5, -0.7, and

-0.8 Pg C $yr^{-2}$ for the 1.2℃, 1.5℃, 2℃, and 3℃ stabilization scenarios respectively, and a maximum change in annual $CO_2$-fe emissions of -0.7, and -1.6 Pg C $yr^{-2}$ for the 2℃ and 3℃ overshoot scenarios (Fig. A1).

## 2.3 Model Evaluation

We briefly summarize previous evaluations of GFDL-ESM2M's physical and biogeochemical state with a focus on the

variables most relevant for this study. In general, the model tends to better simulate physical than biogeochemical variables. The simulated ocean overturning structure and rate, ocean heat transport, ocean temperature and salinity, sea surface height, water mass age distributions and mixed layer depth show a general fair agreement with observational data (Dunne et al., 2012). Some potentially relevant mean biases include an overall too deep thermocline, Antarctic Bottom Water formation through open ocean convection, weak deep Pacific ventilation and too strong North Pacific thermocline ventilation (Dunne et al., 2012).

Furthermore, GFDL-ESM2M performs within the CMIP5 range with respect to $O_2$ projections, despite the global mean $O_2$ concentration being 5% lower than observed due to comparably large low-$O_2$ water volumes (Bopp et al., 2013). The spatial present-day distribution of $O_2$ agrees well with observations, and most of the bias in $O_2$ is limited to the Oxygen Minimum Zones in the eastern equatorial Pacific at depth (Frölicher et al., 2020), similar to other Earth system models (Cabré et al., 2015). The globally integrated NPP is 74 PgC $yr^{-1}$ in GFDL-ESM2M, compared to 53 Pg C $yr^{-1}$ on average in the observation-

based estimates (Le Grix et al. 2022). GFDL-ESM2M overestimates NPP, especially in the low latitudes, but despite these differences, GFDL-ESM2M succeeds in simulating the NPP mean spatial pattern of higher values in the low latitudes and lower values in the Southern Ocean and the subtropical gyres. As a result of the mean temperature and oxygen biases in the model, all mean biases have been corrected for by subtracting the mean bias between World Ocean Atlas 2018 data and the model from all model data (see below).


A comparison of the relative changes in Aerobic Growth Index ($AGI_{rel}$; see section 2.5) between CMIP6 model simulations and GFDL-ESM2M at transient 1.5℃ warming allows additional evaluation of GFDL-ESM2M's performance. In CMIP6 models (Morée et al., 2023), the multi-model mean $AGI_{rel}$ changes are slightly smaller than in the GFDL-ESM2M, while patterns are generally similar. We attribute part of the relatively weaker signal in the CMIP6 ensemble compared to GFDL-

ESM2M to the later timing of reaching (transient) 1.5℃ warming in the GFDL-ESM2M. Nevertheless, pelagic $AGI_{rel}$ largely falls within the CMIP6 range, while demersal $AGI_{rel}$ may be of larger magnitude than CMIP6.

## 2.4 Model Data and Analysis

The annual mean GFDL-ESM2M data used in this study consist of dissolved $O_2$ concentration, potential temperature, salinity,

ideal age, and the particulate organic carbon flux at 100m depth. The model data were first drift corrected by subtracting a



linear trend obtained from the corresponding preindustrial control simulation over 1861-2500. The simulated $O_2$, temperature and salinity data were bias-corrected using World Ocean Atlas observational data (WOA18; Garcia et al., 2019; Locarnini et al., 2018; Zweng et al., 2019) as follows: Climatological mean WOA18 data were regridded horizontally and vertically to the model grid after which the bias between the WOA18 data and the time mean model data over the years 1995-2014 were

subtracted from the full time series. In-situ temperature is calculated from the potential temperature, and $pO_2$ is calculated from in-situ temperature and the salinity and $O_2$ data (Bittig et al., 2018; Morée et al., 2023) .

The 'first year of stable warming level' of its respective global warming level refers to the year in which the 31-yr running mean of the global mean 2-m air temperature is within one standard deviation of the aimed-for warming level (standard

deviation of 0.074°C obtained from the preindustrial control simulation). The 31-yr mean was chosen to exclude interannual-to-decadal natural variability to a large degree. To make a fair comparison between the temperature stabilization scenarios with respect to the maximum amount of equilibration time available after reaching stable warming, committed habitat change is the change in the 296 years after hitting the respective stable warming level. Because the 3°C stabilization simulation reaches its warming level the latest, it determines this 296-year timescale of committed changes. Habitat change at a certain year is

expressed as the forward 31-year mean at that year.

For the species-specific results, we estimate the combined uncertainty (e.g., whiskers in Fig. 3) from the timing of first reaching a stable warming level (varied by ±10 years) and the species-specific control simulation variations in subcritical habitat volume, which are combined using the Root Sum Square of the respective standard deviations. For the overshoot data, only

the control standard deviation is considered as there is no stable warming level hit year.

**2.5 Aerobic Growth Index**

For each species i, the Aerobic Growth Index (AGI) is calculated as the ratio between the $pO_2$ supply and demand (Eq. 1) (Clarke et al., 2021; Morée et al., 2023):

$$AGI_i = \frac{pO_2^{supply}}{pO_{2,i}^{demand}} = \frac{pO_2^{supply}}{pO_{2,i}^{threshold} \cdot (\frac{1}{3})^{1-d} \cdot exp\left(\frac{j_2 - j_1}{T_i^{pref}} \frac{j_2 - j_1}{T}\right)} , \qquad (1)$$

with partial pressure $pO_2$ (units in Pa) and in-situ temperature $T$ (K). The generalized temperature dependence is represented by the variables $j_1$ (the anabolism activation energy divided by the Boltzmann constant, 4500K) and $j_2$ (the catabolism activation energy divided by the Boltzmann constant, 8000K), scaled by the metabolic scaling coefficient $d$ (0.7). Aerobic scope limits the species' distribution. Therefore, by using biogeographical data, we inferred the oxygen threshold necessary to support a viable population of the species. The species-specific critical threshold of $pO_2$ ($pO_{2,i}^{threshold}$) is calculated as the

volume-weighted 10th percentile of 1995-2014 time-mean $pO_2$ within the 3-dimensional distribution of the species. The





estimated $pO_2$ threshold for growth is proportional to critical $pO_2$ measured from physiological experiments (Clarke et al. 2021). The preferred in-situ temperature ($T_i^{pref}$) is calculated as the volume-weighted 50th percentile of the 1995-2014 time-mean temperature within the 3-dimensional distribution of the species, and $AGI_i^{crit}$ as the volume-weighted 10th percentile of 1995-2014 time-mean AGI (Clarke et al., 2021; Morée et al., 2023).


Using $AGI_i$ and $AGI_i^{crit}$, we quantify the percentage of contemporary habitat available to sustain a viable population of the species (i.e., where $AGI_i > AGI_i^{crit}$). Our use of this extended version of AGI (Morée et al., 2023) includes vertical variability in $O_2$ and temperature in both the estimate of species' $pO_{2,i}^{threshold}$, $T_i^{pref}$ and hence $AGI_i$ and $AGI_i^{crit}$, as well as considering fully 3-dimensional changes in $O_2$ and temperature (as opposed to surface or sea floor values only). The generalized temperature dependence of $pO_{2,i}^{demand}$ may cause over- or underestimation of an individual species' response to temperature. The sensitivity of loss of contemporary habitat to this parameterization ($j_2$-$j_1$ in Eq. 1) is minor (Morée et al., 2023).

Changes in AGI between a time $t_1$ and $t_0$, relative to $t_0$ are species-independent and provide a sense of direction and magnitude of change in habitat viability (Morée et al., 2023):

$$AGI^{rel} = \frac{\Delta AGI}{AGI(t_0)} = \frac{pO_2(t_1)}{pO_2(t_0)} \cdot exp\left( (j_2 - j_1) \cdot \left( \frac{1}{T(t_1)} - \frac{1}{T(t_0)} \right) \right) - 1 \qquad (2)$$

The $t_0$ refers to the mean over the period 1861-1900 in this study. Note that small changes in $pO_2$ can lead to large changes in $AGI_{rel}$ if reference $pO_2$ (i.e. $pO_2$ at t=0) is low (Eq. 2). Note that relative changes in AGI can be a poor indicator of the magnitude of change in contemporary habitat volume of specific species, for which species-specific critical thresholds are needed (Morée et al., 2023).

**2.6 Species data**

Spatial distribution data for the 46 representative exploited species (Morée et al., 2023; Palomares et al., 2004) are used to calculate $T_i^{pref}$, $pO_{2,i}^{threshold}$ and $AGI_i^{crit}$. These data also form the reference habitat for assessing changes in contemporary habitat volume. The species were selected such that they provide a representative range in body size, climatic zone, habitat size, and depth range. The 46 species also cover a broad range of vulnerabilities to warming and deoxygenation, with the most vulnerable species having a ~30 times larger change in volume per unit change in AGI than the least vulnerable species (Morée et al., 2023). We include 23 species with their predominant occurrence in the epipelagic (0-200m depth), 5 species that mostly inhabit the mesopelagic (200-1000m depth) as well as 18 demersal species which live on or just above the sea floor (for which we use the deepest ocean model layer). We assess contemporary habitat volume change by extending the 2D habitats over their depth range assuming that the distribution is the same throughout the water column.






All habitat changes are changes in contemporary habitat volume since the 1861-1900 mean, to be consistent with the atmospheric temperature anomalies, as well as being presented as a percentage of viable contemporary habitat volume to facilitate comparison between species.

### 2.7 Adaptation

Species will likely acclimatize and/or adapt to global warming by improving species' temperature and $O_2$ stress resistance on the timescales considered in this study (Pinsky et al., 2020). Such acclimatization and adaptation to environmental changes is already observed over the last few decades, for example for warm-water corals (Lachs et al., 2023; Logan et al., 2021). We base our adaptability assessment on a running climatology approach, which allows for the evaluation of both adaptation pressure to a species' habitat and the timescale of a species' adaptation to such pressure. We thereby follow the approach

applied to corals by Logan et al., (2014).

Adaptation pressure is evaluated by considering for each species a time-dependent $AGI_i^{crit}$ ($AGI_{i,t}^{crit}$) that changes when in-habitat AGI (and hence $pO_2$ and/or temperature) changes. As $AGI_i$ and thus $AGI_{i,t}^{crit}$ depends on $T_i^{pref}$, $pO_{2,i}^{threshold}$, we also calculate a time-dependent $pO_{2,i}^{threshold}$ ($pO_{2,i,t}^{threshold}$) and $T_i^{pref}$ ($T_{i,t}^{pref}$). We do so by following the standard approach

(described in section 2.5) but instead using a moving 20-year time-mean time window of $pO_2$, temperature and AGI before each year t. In this way, $AGI_{i,t}^{crit}$ is calculated for each species and each year over the period 1861 to 2500. It also follows that $AGI_{i,t=2014}^{crit}$ is equal to the standard $AGI_i^{crit}$. For the first 20 years of our dataset, we use the time mean of all years available before and including that particular year.

We then consider the timescale of adaptation and its uncertainty by using mean $AGI_{i,t}^{crit}$ over the 40, 60, 80, 100 (the range in Logan et al., 2014), as well as 140, 180, and 220 years (to extend to species with longer adaptation times due to, for example, slower generational overturn) before year t in our quantification of contemporary habitat volume change at year t. A 40-year adaptation timescale thus means that a species has adjusted (i.e., adapted) its $AGI_{i,t}^{crit}$ to the mean $AGI_{i,t}^{crit}$ over the previous 40 years.

## 3 Methods

### 3.1 Committed loss of contemporary habitat and overshoot impacts

In the stabilization simulations, the global atmospheric surface warming levels of 1.2℃, 1.5℃, 2℃, and 3℃ are reached in the years 2019, 2054, 2091, and 2162 respectively (Fig. 1a). Afterwards, atmospheric temperatures are stabilized. Concurrently, the ocean is projected to warm and lose oxygen (Fig. 1c,e). By 2400-2500, the habitats of the 46 representative

marine species have warmed by a median 0.8℃, 1.1℃, 1.5℃ and 2.2℃ (Fig. 1c), and lost 1.2 mbar, 1.3 mbar, 1.8 mbar, and





3.1 mbar of $p$O$_2$ in 2400-2500 for the 1.2℃, 1.5℃, 2℃, and 3℃ stabilization scenarios, respectively (median across all species; Fig. 1e).

**Figure 1: Simulated temperature stabilization and overshoot scenarios and associated impacts on median in-habitat temperature and $p$O$_2$ changes as well as on contemporary habitat volume across 46 representative marine species.** (a,b)





Global mean 2-m air temperature change since 1861-1900 for (a) temperature stabilization scenarios at 1.2℃, 1.5℃, 2℃, and 3℃ warming, as well as (b) overshoot scenarios that peak at 2°C and 3°C warming before returning to 1.5°C warming. (c-h) Median change in (c,d) in-situ temperature and (e,f) $pO_2$ in the respective habitats of the 46 species and in contemporary habitat

volume (% of volume) since 1861-1900 for the (c,e,g) stabilization and (d,f,h) overshoot scenarios. Whiskers in (c-h) indicate the 25th to 75th percentile range across the species habitat at the respective year of starting temperature stabilization and in year 2500. Thick lines show the median across the individual species' 31-year running mean data and thin lines the median across the species' their annual mean time series.

With habitat warming and deoxygenation, species' habitat volume, defined here as the volume of contemporary habitat with
the Aerobic Growth Index (AGI) above a species-specific threshold (see Methods), shrinks across the 46 species under all stabilization scenarios (Fig. 1g). Contemporary habitat volume continues to decrease even after atmospheric temperatures have stabilized. As a result, only about half of the total loss of contemporary habitat volume is realized when first reaching the respective stable warming levels: The median fraction of realized habitat loss upon hitting stable warming levels, compared to the loss 296 years thereafter, amounts to 0.45 (with 25th-75th percentiles across the species at 0.19-0.76) for the 1.2℃ scenario,
0.41 (0.25-0.67) for 1.5℃, 0.67 (0.36-0.93) for 2℃, and 0.60 (0.40-0.81) for the 3℃-stabilization scenario (Table 1). Adverse impacts on habitat viability of marine species thus continue to aggravate for centuries despite stabilization of the global mean atmospheric temperature.

**Table 1. Simulated median habitat change (ΔHabitat) across the 46 representative species, presented in % of**
**contemporary habitat volume at (a) time of warming level hit and (b) 296 years after time of warming level hit. (c)**
**Difference between (a) and (b), and (d) ratio between (a) and (b).** In all columns the 25th and 75th percentile across the species is indicated in brackets and all values are 31-year forward running means.

| | *ΔHabitat (%)* | | | |
| --- | --- | --- | --- | --- |
| *Warming level (°C)* | *a. At time of warming level hit* | *b. At time of warming level hit + 296 years* | *c. Committed (=b-a)* | *d. Ratio (=a/b)* |
| *1.2* | **-0.48 (-0.96, -0.34)** | **-1.06 (-2.32, -0.89)** | **-0.58 (-1.48, -0.28)** | **0.45 (0.19, 0.76)** |
| *1.5* | **-0.76 (-1.32, -0.58)** | **-1.83 (-2.73, -1.31)** | **-1.07 (-1.72, -0.55)** | **0.41 (0.25, 0.67)** |
| *2* | **-1.53 (-2.81, -1.15)** | **-2.28 (-3.92, -1.75)** | **-0.75 (-2.03, -0.06)** | **0.67 (0.36, 0.93)** |
| *3* | **-2.78 (-4.40, -2.24)** | **-4.59 (-7.13, -3.14)** | **-1.82 (-2.94, -0.55)** | **0.60 (0.40, 0.81)** |



After 296 years of temperature stabilization, the median decline in contemporary habitat among the species ranges from 1.06% (with 25th-75th percentiles across the species at 2.32% to 0.89%) for the 1.2°C stabilization scenario to 4.59% (7.13% to 3.14%) for the 3°C-stabilization scenario (Fig. 1g; Table 1). While these changes may appear small, the corresponding volumetric losses are substantial, ranging from thousands (order for the demersal species) to millions of km³ (order for the epipelagic species), depending on the species and scenario (Morée et al., 2023). Notably, the 25% of species with the largest losses

experience 2-5 times larger losses depending on the species and warming level (whiskers in Fig. 1g).

Temporarily overshooting the 1.5°C warming level to 2°C or 3°C (Fig. 1b) drives bigger loss of contemporary habitat volume than direct stabilization at 1.5°C (Fig. 1h), due to associated overshoots of in-habitat warming (Fig. 1d) and deoxygenation (Fig. 1f). Peak loss of contemporary habitat volume during the 2°C and 3°C overshoot scenario is -2.1% and -3.2% in year

2270 and 2246, respectively (medians across the species, Table 2). This corresponds to -0.5% and -1.7% more loss of contemporary habitat than under the 1.5°C stabilization scenario at that time. Notably, peak loss of contemporary habitat is realized 166 years (for the 2°C overshoot scenario) and 66 years (3°C overshoot) after maximum atmospheric warming in year 2104 and 2180 (Fig. 1b,h). By the time the maximum median loss of contemporary habitat is realized, atmospheric warming has therefore already returned to 1.7°C and 2.5°C, respectively. When atmospheric warming has returned to 1.5°C in 2400-

2500, median contemporary habitats are 0.06% larger (0.31% smaller) in the 2°C (3°C) overshoot scenario than in 1.5°C stabilization scenario. This indicates that for most species overshoot impacts are largely reversible by this time (Table 2). Nevertheless, the strong adverse impacts during the overshoot underscore the advantages of stabilizing at 1.5°C warming over overshooting this level.

**Table 2. Simulated median habitat change (ΔHabitat) across the 46 representative species, presented in % of**
**contemporary habitat volume at (a) the year of maximum change for the respective scenario, (b) in 2400-2500, (c) the difference with the 1.5°C stabilization scenario in 2400-2500, and (d) the ratio relative to the 1.5°C stabilization scenario in 2400-2500.** In (d), the overshoot legacy is the difference between the overshoot scenario and the 1.5°C warming scenario relative to the change in the 1.5°C warming scenario. In all columns the 25th and 75th percentile across the species is indicated in brackets and all values are 31-year forward running means.


| | ΔHabitat (%) | | | |
|---|---|---|---|---|
| *Scenario* | *a. At peak change* | *b. In 2400-2500* | *c. Overshoot legacy in 2400-2500 minus stable 1.5°C* | *d. Overshoot legacy ratio in 2400-2500 relative to stable 1.5°C* |
| *2°C overshoot* | -2.13 (-3.11, -1.60) | -1.75 (-3.12, -1.24) | 0.06 (-0.37, -0.03) | 0.03 (-0.15, -0.03) |



| | | | | |
|---|---|---|---|---|
| *3°C overshoot* | -3.16 (-4.01, -2.42) | -2.12 (-3.66, -1.31) | -0.31 (-0.92, -0.10) | -0.17 (-0.36, -0.07) |

## 3.2 Spatial patterns and drives of habitat viability changes

To generalize these median impacts beyond the 46 analyzed species and to investigate the drivers of changes, we calculate the local relative changes in habitat viability expressed as the changes in the Aerobic Growth Index relative to the period 1861-
1900 (AGI$^{rel}$). These relative changes in habitat viability are fully species-independent and indicate the direction and extent of change in habitat viability across the world's oceans. We focus on the 1.5°C stabilization and the 2°C overshoot scenario, but results are qualitatively similar for the other scenarios.

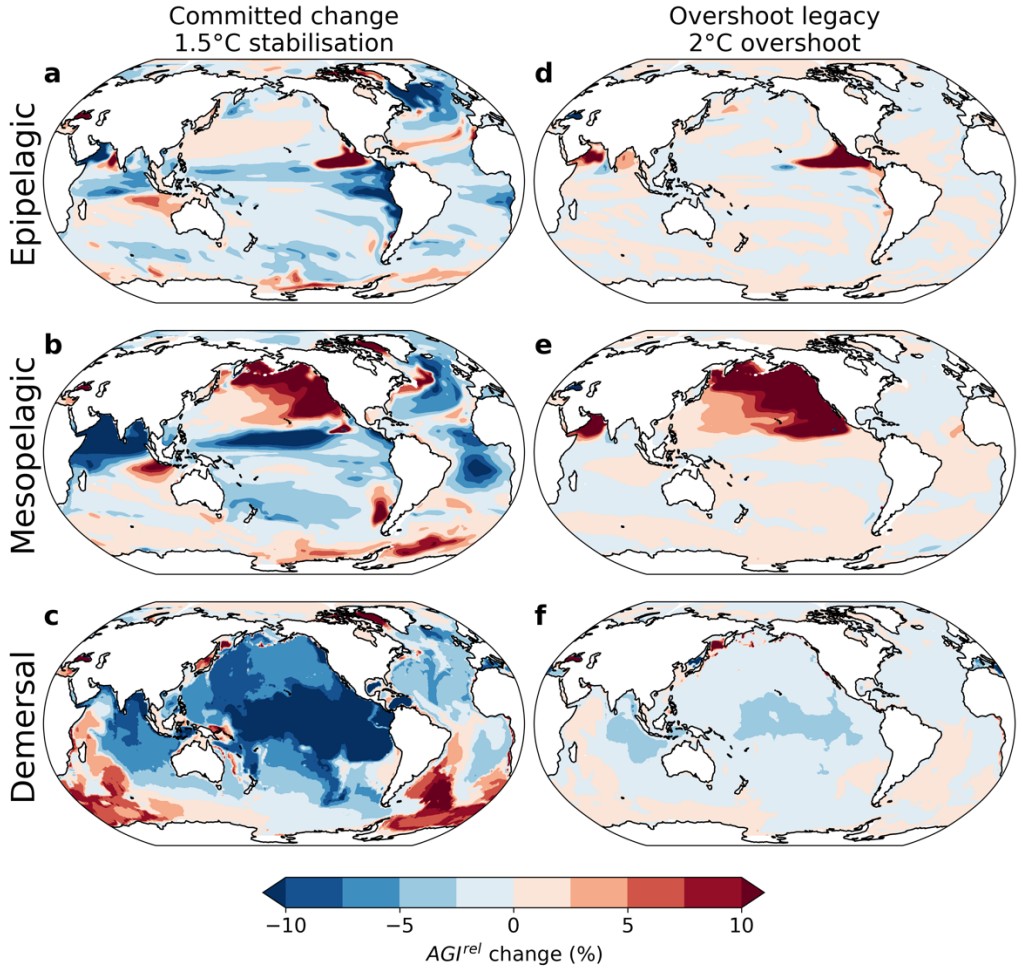

**Figure 2. Simulated committed change in the 1.5°C stabilization scenario (left panels) and overshoot legacy in the**
**2°C overshoot scenarios (right) in relative changes in habitat viability for three depth realms. (a-c)** Committed change



in habitat viability is expressed as the difference between AGI$^{rel}$ at 269 years after temperatures have stabilized and when the 1.5°C warming level is hit (i.e., the committed period). (**d-f**) Overshoot legacy in habitat viability is expressed as the difference between AGI$^{rel}$ in the 2°C overshoot and the 1.5°C stabilization scenarios in 2400-2500. Results are shown as the vertical mean over the epipelagic (0-200m) and mesopelagic (200-1000m) realms, and the demersal (sea floor) realm.


In the 1.5°C stabilization scenario, committed decreases in AGI$^{rel}$ (i.e., after temperatures have stabilized) intensify with depth and are comparable in magnitude to the changes during the period until 1.5°C warming is first reached (Fig. 2a-c). The most substantial committed declines in the epipelagic realm occur in the North Atlantic (Fig. 2a), driven by a sustained rise in ocean temperature (Fig. 3a) linked to the recovery of the Atlantic Meridional Overturning Circulation during the stabilization phase

(Frölicher et al., 2020, Lacroix et al. 2024). Elsewhere in the pelagic and demersal realms, committed changes in habitat viability are predominantly driven by $O_2$ changes through changes in apparent oxygen utilization (i.e., changes in ocean ventilation and organic matter remineralization) (Figs. 3 and A2). In the Pacific demersal realm (Fig. 2c), there's a committed reduction in habitat viability due to $O_2$ decrease tied to reduced deep ocean ventilation as indicated by older water ages (Fig. 3). Conversely, there is a committed increase in habitat viability in the North Pacific mesopelagic realm and Southern Ocean

mesopelagic and demersal realms. The North Pacific increase is linked to increased ventilation and reduced biological oxygen consumption (Figs. 2 and A2). The Southern Ocean increase is mainly driven by enhanced ventilation as indicated by younger water ages (Fig. 2).







**Fig. 3. Simulated committed changes in the environmental state for the 1.5°C stabilization scenario for three depth realms.** (a-c)
Committed change in in-situ temperature, (d-f) $pO_2$, (g-i) $AGI^{rel}$ due to changes in $O_2$ only (calculated by keeping temperature constant at
its 1861-1900 mean), (j-l) Apparent Oxygen Utilization (AOU=$O_2^{sat}$-$O_2$), and (m-o) ideal age of the water. As a proxy for change in
ventilation time, we used the ideal age tracer, which is set to zero in the surface ocean and ages at a rate 1 yr yr$^{-1}$ below that. Committed
change is the difference between environmental state at 269 years after temperatures have stabilized and when the 1.5°C warming level is
hit. Results are shown as the vertical mean over the epipelagic (0-200m), mesopelagic (200-1000m), and demersal (sea floor) realms.


The overshoot legacy for the 2°C overshoot is determined by comparing the changes in the overshoot scenario with the 1.5°C
stabilization scenario without overshoot in the period 2400-2500, when global warming levels are similar (Fig. 2d-f). The most
pronounced overshoot legacies are simulated in the northern flank of the eastern equatorial Pacific epipelagic (Fig. 2d) and in
the mesopelagic of the North Pacific (Fig. 2e). These regions of habitat viability increase are caused by higher $O_2$ levels
resulting from reduced $O_2$ consumption and rejuvenating mesopelagic waters (Fig. 4j,k), which is consistent with previous





idealized model simulations (Li et al., 2020; Santana-Falcón et al., 2023). In the demersal realm, relative changes in habitat viability are generally more negative under the overshoot than in the stabilization scenario by 2400-2500 (Fig. 2f). This negative legacy is mainly driven by reduced deep ocean ventilation as indicated by increased water ages (Fig. 4).



**Fig. 4. Simulated overshoot legacy for the 2°C overshoot in the environmental state for three depth realms in 2400-2500.** (a-c) Overshoot legacy in in-situ temperature, (d-f) $pO_2$, (g-i) $AGI^{rel}$ due to changes in $O_2$ only (calculated by keeping temperature constant at its 1861-1900 mean), (j-l) Apparent Oxygen Utilization ($AOU = O_2^{sat} - O_2$), and (m-o) ideal age of the water. As a proxy for change in ventilation time, we used the ideal age tracer, which is set to zero in the surface ocean and ages at a rate 1 yr yr$^{-1}$ below that. Overshoot legacy is the difference between state in the overshoot scenario and 1.5°C stabilization scenario in 2400-2500. Results are shown as the vertical mean over the epipelagic (0-200m), mesopelagic (200-1000m), and demersal (sea floor) realms.





### 3.3 Impacts on contemporary habitat volume of individual species

The committed loss of contemporary habitat volume is generally largest for demersal species (median of 1.7% in the 1.5°C
stabilization scenario, and up to 6.4% for *Spectrunculus grandis*) and typically smallest in the epipelagic realm (median of
0.7%) (grey bars in Fig. 5a). When reaching stable 1.5°C warming in 2054, only a median 47% of the total loss is realized in
the epipelagic, 56% in the mesopelagic and 28% in the demersal realm (Fig. A3a). Both the committed loss of contemporary
habitat volume and the contrast in impact between the different depth realms generally strengthen with increased levels of
global warming (Fig. A4). For most species, the committed loss of contemporary habitat volume is driven by ongoing decreases
in dissolved $O_2$ (black stars in Fig. 5a). However, the magnitude of loss of contemporary habitat for individual species also
depends on species-specific vulnerability and not only on the magnitude of changes in AGI[rel], temperature or $O_2$ (Morée et al.,
2023).

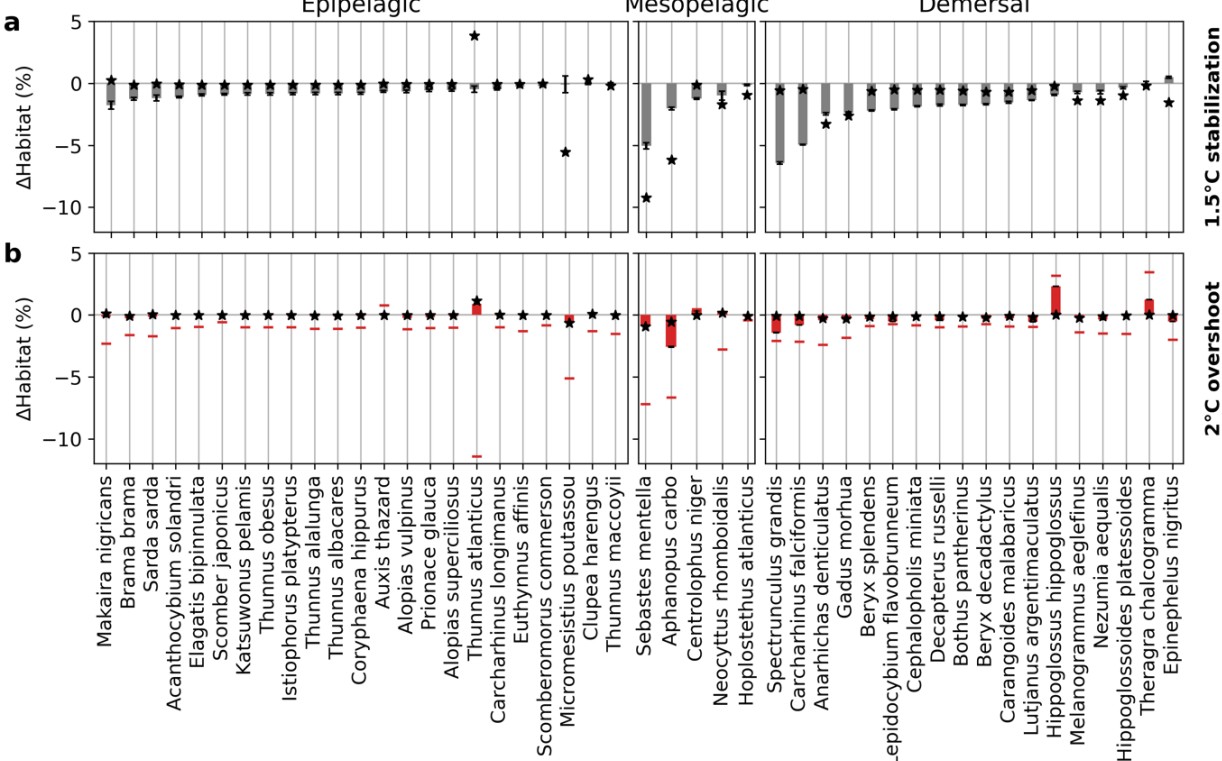

**Figure 5. Simulated committed changes in contemporary habitat volume for the 1.5°C stabilization scenario (top panels) and
overshoot legacy in the 2°C overshoot scenario (bottom) for 46 representative marine species and three different depth realms.** (a)
Committed change (grey bars) expressed as the difference between contemporary habitat volume changes at 269 years after temperatures
have stabilized and when the 1.5°C warming level is hit (i.e., the committed period). (b) Overshoot legacy (red bars) expressed
as the difference between the 2°C overshoot scenario and the 1.5°C stabilization scenario in 2400-2500. Black stars in (a,b) indicate



contemporary habitat changes that are driven by temperature changes only (i.e., keeping $O_2$ values at 1861-1900 conditions). Maximum changes during the overshoot are indicated with red horizontal bars in (b). Whiskers in (a) indicate the uncertainty as the combined uncertainty coming from the uncertainty in timing of warming level hit and the species-specific control simulation variability (in (b), only the species-specific control simulation variability is considered in the whiskers as no stable warming level hit year is considered; see Methods).

When temporarily overshooting stable 1.5°C warming to 2°C, median peak overshoot habitat loss exceeds the impact of 1.5°C warming by 1.1%, and at most 11.4% for *Thunnus atlanticus* (bars in Fig. 5b; see Fig. A4 for the 3°C overshoot). After overshoot returns in 2400-2500 from 2°C to stable 1.5°C warming, the species contemporary habitat volume is between -2.6% and +2.3% of stable 1.5°C warming (Fig. 5b), with a median loss of contemporary habitat volume of 0.10% in the epipelagic, 0.11% in the mesopelagic and 0.36% in the demersal realm (medians across depth realms in Fig. 5b). The overshoot legacy is
predominantly explained by enhanced (de)oxygenation in the overshoot simulation relative to the stabilization simulation in 2400-2500 (stars in Fig. 5b). Some species like *thunnus atlanticus* and *hippoglossus hippoglossus* even benefit from overshooting 1.5°C warming, mainly due to local oxygenation relative to the 1.5°C stabilization scenario.

**3.4 Potential adaptation**

We consider both the adaptation pressure to a species which would incite adaptation (as the in-habitat change in habitat viability quantified by AGI) as well as the timescale of adaptation (the number of years needed to adapt a species' thresholds, evaluated for 40, 60, 80, 100, 140, 180, and 220 years). In consequence, after being under high (low) adaptation pressure a species will be relatively insensitive (sensitive) for a time that increases with the timescale of adaptation.

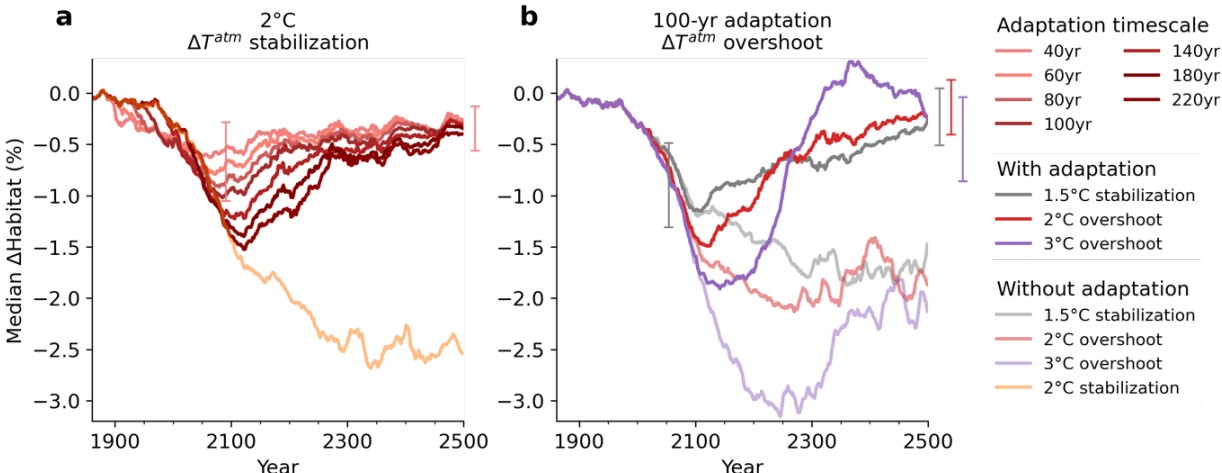

**Figure 6: The impact of potential adaptation on simulated change in contemporary habitat volume across 46 representative marine species in the 2°C stabilization (left panel) and 2°C and 3°C overshoot (right) scenario. (a)**





Median habitat change since 1861-1900 for different adaptation timescales and the non-adaptive 2°C stabilization scenario. (**b**) Median habitat change since 1861-1900 for the 2°C and 3°C overshoot scenarios and the 1.5°C stabilization scenario assuming a 100-year adaptation timescale, as well as their non-adaptive counterparts. In both panels, whiskers indicate the 365 25[th] and 75[th] percentile limits across the species for the 40-year adaptation timescale in year 2091 when first reaching the 2°C warming level and in year 2500. Timeseries are smoothed with a 31-yr running mean.

Adaptation substantially reduces the projected loss of contemporary habitat compared to non-adaptation (Fig. 6). In the 2°C-stabilization scenario that includes adaptation, loss of contemporary habitat temporarily peaks at 0.63-1.53% around 2100 370 depending on the timescale of adaptation, before gradually recovering to 0.26-0.40% loss by 2500 (Fig. 6a, qualitatively similar results are obtained for the other scenarios). In comparison, the loss of contemporary habitat without adaptation is 2.54% by 2500. The maximum loss of contemporary habitat is larger and later for longer adaptation timescales. However, after the 21[st] century, loss of contemporary habitat in the adaptation scenarios is reduced again, because critical thresholds in AGI have had time to decrease enough (i.e., species could adapt to tolerate less favorable conditions).

In the overshoot scenarios, adaptation also causes reduced impact when compared to the non-adaptive scenarios (Fig. 6b). By 2400-2500, adaptation reduces the 3°C (2°C) overshoot legacy in terms of loss of contemporary habitat to 0.04% (0.28%), while this is 2.11% (1.75%) without adaptation. Following our approach, we find that the stronger the overshoot a species is exposed to, the lower the species' critical threshold becomes through adaptation (i.e., adapting to lower AGI[crit] and hence becoming more tolerant to low-oxygen or high-temperature environments). This results in a 'double overshoot' in the 3°C-380 overshoot simulation, where habitat volume impacts after ~2250 are smaller than in an adaptive 1.5°C stabilization scenario because species are still adapted to the (much) less favorable conditions in this recovery phase (Fig. 6b). In all scenarios with adaptation, median impact on contemporary habitat volume does not reduce to zero because some species are exposed to a continued decrease in habitat viability even at the end of the 25[th] century.

## 4 Discussion and Conclusions

Through novel policy-relevant stabilization scenarios using an Earth system model, we show that only about half of the fish habitat changes have been realized when the temperature stabilization level is first hit. In overshoot scenarios, it may take over 150 years after the peak temperature for the most substantial impacts on marine species to manifest. Additionally, our findings suggest that rapid adaptation to changing conditions by species could mitigate some of the effects, contingent upon the rate of adaptation and external pressure.

An important assumption in our analysis is that the employed single Earth system model GFDL-ESM2M simulates changes in temperature and dissolved $O_2$ in a sufficiently realistic manner. We consider our global analysis for the warming and deoxygenation robust, especially given the good agreement of the model with observations and with other Earth system models



(see Methods). However, projected local changes in oxygen may be less robust (Kwiatkowski et al., 2020), especially in low $O_2$ zones, and are likely to be underestimated (Buchanan & Tagliabue, 2021). Additional ESMs should be run under policy-relevant temperature stabilization and overshoot scenarios to quantify the uncertainty in $pO_2$ changes. Nevertheless, multiple modelling studies indicate a centennial to multimillennial response timescale of $O_2$ to global warming (Battaglia & Joos, 2018; Bertini & Tjiputra, 2022; Frölicher et al., 2020), making it likely that multi-centennial $O_2$ changes are indeed important for committed impacts and overshoot legacies. Therefore, our foremost conclusion, that a major part of the total loss of contemporary habitat volume occurs after global temperatures have stabilized, likely does not depend on the model used - but the quantitative results may vary when using other models.

We highlight a multi-decadal to centennial delay in peak impact in contrast to peak global warming in the overshoot scenarios. This is much longer than the eight year delay found in Meyer et al., (2022) for marine species in a 2°C overshoot. Their use of sea surface data likely causes an underestimation of the peak delay due to faster reversibility of temperature and $O_2$ at the surface as compared to depth (Santana-Falcón et al., 2023; Schwinger et al., 2022), providing an additional argument for using 3-dimensional species distributions (GA, 2017). The depth of a species' distribution is also central to the quantification of the legacy impact of an overshoot after return to stable 1.5°C warming, since reversibility timescales increase with depth (Santana-Falcón et al., 2023; Schwinger et al., 2022). Furthermore, the steep vertical gradients of $pO_2$ and temperature as well as the dampened climate change signal at depth provide additional arguments to include the third, vertical, dimension when estimating (changes in) contemporary species distributions (GA, 2017).

The Aerobic Growth Index faces limitations in its application due to several factors, notably the absence of species-specific fully three-dimensional distribution data, its neglect of critical stressors beyond warming and deoxygenation, and its confinement to assessing solely the loss of contemporary habitat. While we aim to capture habitat variability in $O_2$ and temperature relevant for species-specific critical thresholds by extrapolating two-dimensional habitats across different depth ranges, these approximations may not fully capture the complete picture. Despite this, our findings align with variable species responses seen in distribution shifts (Poloczanska et al., 2016), emphasizing the potential for substantial alterations in species compositions (Gotelli et al., 2022). Additionally, the response of marine organisms to global warming may encompass sensitivities to other effects, such as changes in low temperatures, acidity, nutrient availability, phenology, disease, predation pressure, invasive species, and (over)fishing (Gissi et al., 2021; Intergovernmental Panel on Climate Change (IPCC), 2023). Multi-stressor research aims to assess and predict the cumulative effects of these stressors, including synergies and antagonistic relationships between them but is still bound by many challenges before full assessments can be made (Gissi et al., 2021; Orr et al., 2020). Last, we note the importance of knowledge of species-specific critical $pO_2$ thresholds and preferred temperature environments. Relative changes in $pO_2$ supply over $pO_2$ demand ratios have been implied to assess impact (Battaglia & Joos, 2018; Deutsch, Ferrel, Seibel, et al., 2015; Oschlies, 2021; Santana-Falcón et al., 2023), but species-specific thresholds and preference windows are needed for such estimates (Morée et al., 2023). Unlike the Metabolic Index, which depends on laboratory estimates of critical thresholds, AGI is widely applicable to all species for which the distributions are known thanks





to its generalized temperature dependence and species' distribution-based critical thresholds. In the future, the application of the AGI to more species will allow the assessment of a wider range of interspecies responses.

Our sensitivity analysis shows that adaptation has large potential to mitigate the impacts of global warming on the habitat viability of marine fishes, although observations of a reduced equatorward extent of species' habitats suggest that adaptation 430 may be too slow, particularly for larger species such as marine fishes (Hastings et al., 2020). Furthermore, we strengthen the notion that the pathway as well as the magnitude of temperature overshoot is important for future ecological impact (Meyer & Trisos, 2023), and additionally show that warming pathways can further modulate impact through their effect on adaptation pressure. However, there is still a large gap in knowledge about marine fishes' adaptation to warming and deoxygenation. Particularly, the rate of adaptation to warming and deoxygenation would depend on many biological and ecological factors 435 that are not considered in this study e.g., the existing genetic diversity, species' life history traits, and population/meta-population structure and connectivity.

We conclude that losses of marine species' contemporary habitats continue for centuries beyond reaching stable global warming levels and after peaking global warming in a temporary overshoot. Any impact assessed at transient warming levels (Hausfather et al., 2022; Morée et al., 2023) thus largely underestimates the total impact on marine species.

440

**Appendix A: Additional figures**



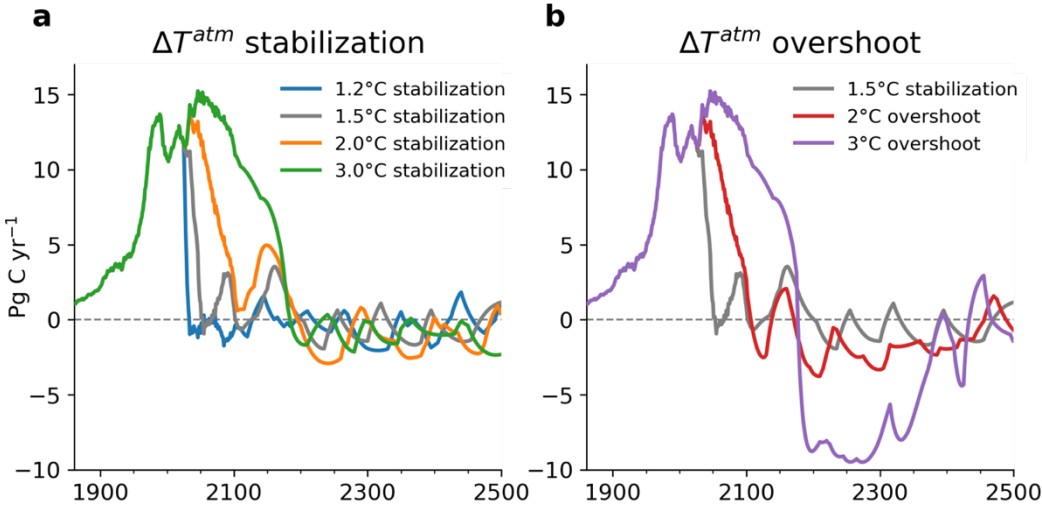

**Fig. A1. Prescribed annual CO₂ forcing equivalent emissions in the GFDL-ESM2M simulations.** (a) CO₂-fe emissions in the temperature stabilization scenarios. (b) CO₂-fe emissions in the temperature overshoot scenarios and the reference 1.5°C stabilization scenario.

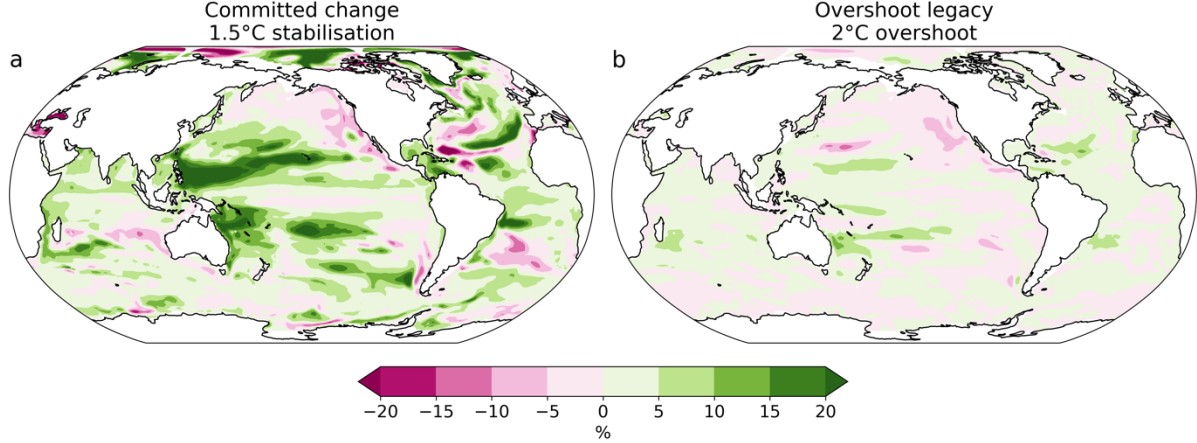

**Fig. A2. Simulated committed changes in the 1.5°C stabilisation (a) and overshoot legacy in the 2°C overshoot scenario (b) of particulate organic carbon export flux at 100m depth.** (a) Committed changes are expressed as the ratio between the change during the committed period (the 269 years after stable warming level hit) and at the time when the 1.5°C warming level is first hit. (b) Overshoot




legacy of the 2°C-overshoot scenario as the difference between the 2°C overshoot and the 1.5°C stabilization scenario divided by the 1.5°C stabilization scenario export flux in 2400-2500.

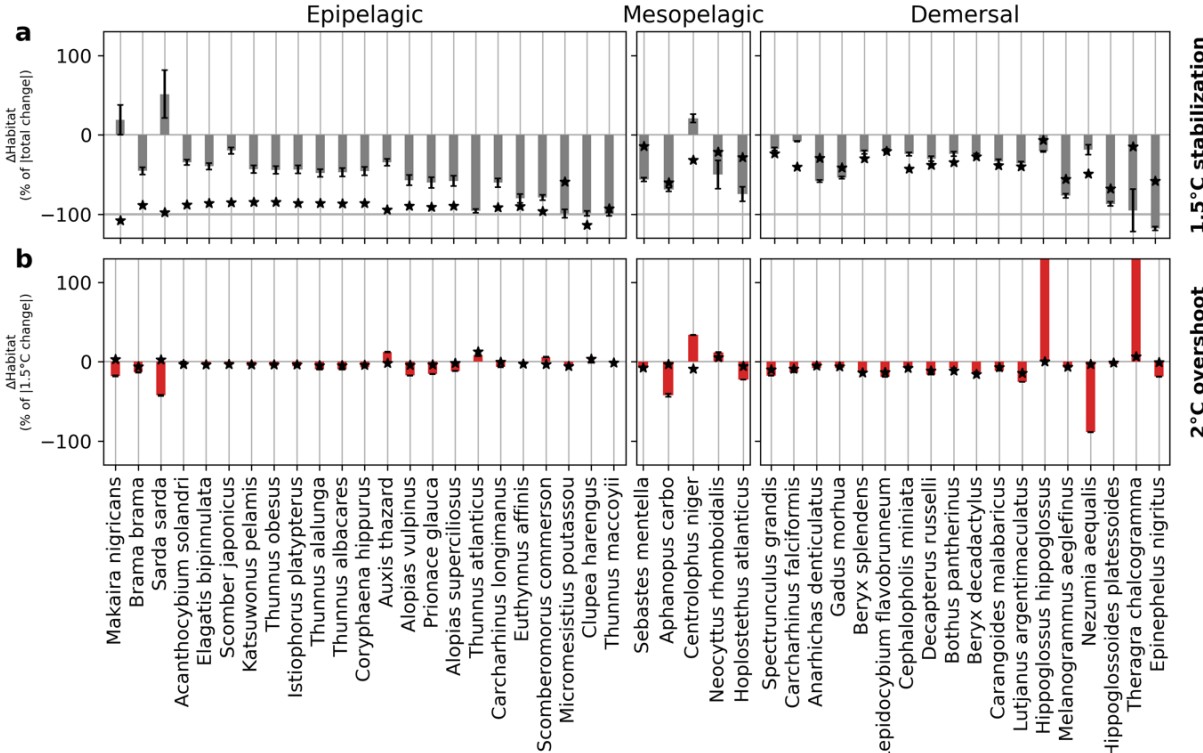

**Fig. A3. Simulated committed changes in the 1.5°C stabilization scenario (top panels) and overshoot legacy in the 2°C overshoot scenario (bottom panels) expressed in relative changes in contemporary habitat volume for 46 representative marine species and for three different depth realms.** (a) Relative committed change as the ratio of habitat change at the time of stable warming level hit and the absolute total change in habitat after stabilization 269 years later. Values of -100% or +100% (-100% is highlighted by a horizontal grey line) indicate that all change is realized at the time of reaching the warming level. (b) Relative overshoot legacy (red bars) is expressed as the ratio between the 2°C overshoot scenario and the 1.5°C stabilization scenario in 2400-2500. *Hippoglossus hippoglossus* and *theragra chalcogramma* have a relative overshoot legacy of 165% and 199%, respectively. Black stars in (a,b) indicate contemporary habitat changes that are driven by temperature changes only (i.e., keeping O₂ values at 1861-1900 conditions; see Methods). Whiskers in (a) indicate the uncertainty as the combined uncertainty coming from the uncertainty in warming level hit timing and the




465 species-specific control simulation variability (in (b), only the species-specific control simulation variability is considered in the whiskers as no warming level hit year is considered; see Methods).

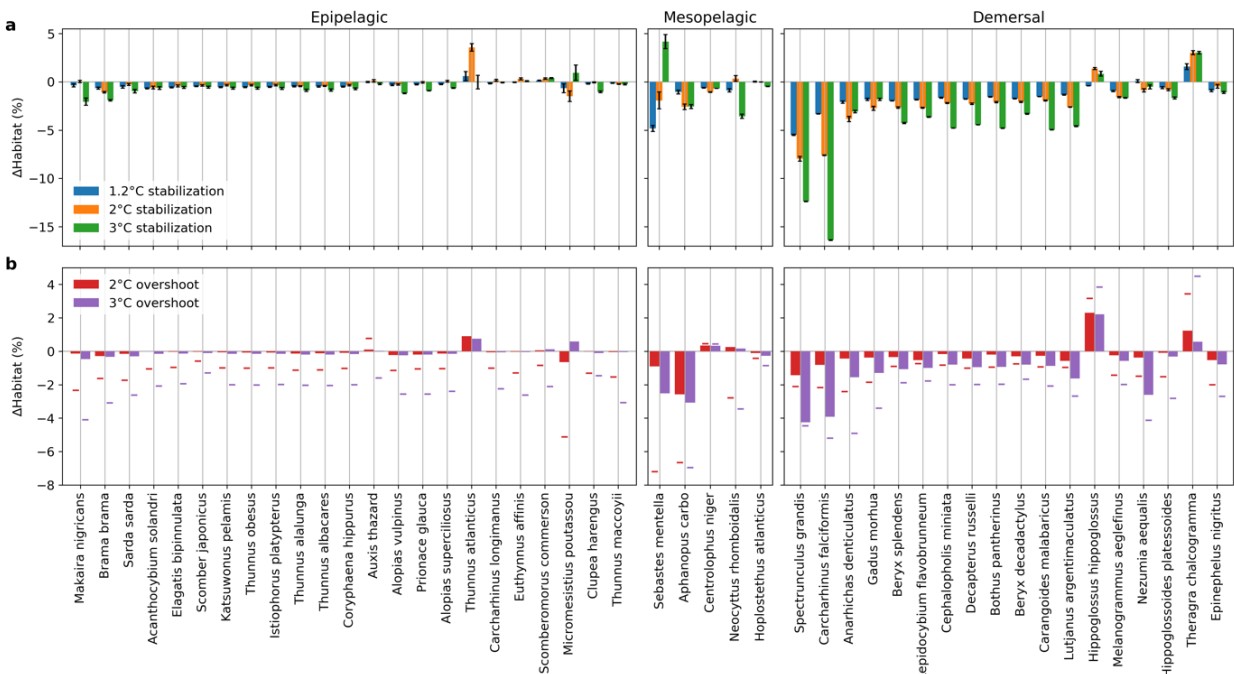

**Fig. A4. Simulated changes in contemporary habitat volume (%) for 46 representative marine species for different stabilization (top panels) and overshoot (bottom) scenarios.** (a) Committed change expressed as the difference between
470 contemporary habitat volume changes at 269 years after temperatures have stabilized and when the 1.5°C warming level is hit (i.e., the committed period). (b) Overshoot legacy as the difference between the 2°C or 3°C overshoot and 1.5°C stabilization scenario in 2400-2500. Maximum changes during the overshoot (10-year running mean filter) are indicated with horizontal bars. Whiskers in (a) indicate the uncertainty as the combined uncertainty coming from the uncertainty in warming level hit timing and the species-specific control simulation variability (in (b), only the species-specific control
475 simulation variability is considered in the whiskers as no warming level hit year is considered; see Methods).

**Code and Data Availability**
480 Data and scripts for reproduction of the manuscript figures are freely accessible at (Zenodo DOI added at acceptance).

**Author contribution**



AM and TF designed the study. FL conducted the Earth system simulations. WWLC was instrumental in developing the Aerobic Growth Index. AM performed the analysis and wrote the first draft. All authors contributed to the writing of the manuscript.

## Competing interests

The authors have not conflicts of interest to declare.

## Acknowledgments

This research was supported by the Swiss National Science Foundation (No. PP00P2_198897) and the European Union's Horizon 2020 research and innovation program grant No. 01003687 (PROVIDE). WWLC was supported by the SSHRC Partnership Grant through the Solving-FCB Partnership. The simulations have been conducted at the Swiss National Supercomputing Centre (CSCS). We thank Tayler Clarke for her contributions to the development of the Aerobic Growth Index, and Jens Terhaar, Fortunat Joos, Mathias Aschwanden, Friedrich Burger and Yona Silvy for their contributions to the development of the Adaptive Emission Reduction Approach. The work reflects only the authors' view; the European Commission and their executive agency are not responsible for any use that may be made of the information the work contains.

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

        Volume 2: Salinity. *NOAA Atlas NESDIS*, *82*(July).