# Peer review of "Long-term impacts of global temperature stabilization and overshoot on exploited marine species"

_EGUsphere, 2024_

## Referee Comment (RC1)

**Review of:**

**"Long-term impacts of global temperature stabilization and overshoot on exploited marine species" by Morée et al.**

Biogeosciences manuscript number egusphere-2024-3090

**Dijon, November 14, 2024**

**Alexandre Pohl** (CNRS Researcher @ Biogéosciences, UMR 6282 CNRS / Université de Bourgogne, 6 Boulevard Gabriel, 21000 Dijon, France)

**Paper summary**:

Morée et al. quantify the impact of ocean warming and deoxygenation on the habitability of the ocean by 46 exploited marine species until year 2500. The authors consider temperature stabilization scenarios and overshoot scenarios, with temperature in the latter case reaching a maximum before decreasing and stabilizing. Ocean habitability is calculated using the aerobic growth index and environmental conditions simulated in transient simulations conducted with the GFDL ocean-atmosphere general circulation model. Results demonstrate that only around half of the total habitat loss is realized when target warming levels are reached. Habitat loss continues in the decades to centuries after that. Species adaptation may lower the overall habitat loss.

**General comment**:

The manuscript represents a welcome departure over previous work. It is clearly written and illustrated with figures of excellent quality that efficiently convey key information supporting the text. The methods require lots of attention and are not always easy to read, but this is for the better since the setup is described in detail and I do not think this aspect can be improved without being detrimental to the scientific content of the manuscript. The results are also very well organized, starting with describing the overall changes in aerobic growth index, then the underlying environmental / climatic drivers, before focusing on changes in habitability at the species level and finishing with an analysis of the impact of adaptation, overall constituting a very informative and comprehensive work. I encourage the publication after very minor revisions and provide suggestions of improvements below.

Please note that I was not able to try and run the code, which was not provided with the manuscript (with a simple placeholder found in the 'code and data availability' section).

**Main comments**:

**1.** Selection of the 46 species. It would be beneficial to provide a short description of the species selected in the Methods: what are these species, and how were they selected? A reader being not familiar with previous work cited would better understand the context of the study.

**2.** Selection of the climatic scenarios. Similarly, it may be instructive to tell a bit more about the choice of the stabilization scenarios. How were the magnitude and time scales of temperature change selected? It may be good to discuss these numbers with regards to the IPCC scenarios. Regarding the overshoot scenarios, it may be interesting to discuss the plausibility of the time scale

of overshoot and subsequent stabilization: is the time scale plausible, based on previous work that did not force temperature changes but mechanistically simulated them, e.g. using sediment-enabled earth system models of intermediate complexity? Such discussion would better frame the context of the work and hence strengthen the study.

**3.** Infinite migration potential. An implicit assumption of the approach is that studied species can always populate the regions of the ocean where oxygen supply is sufficient for them. In reality, we might imagine that some species would not be able to keep up with changing environmental conditions due to limited migration potentials, which may impact the calculated habitability and reversibility and, in some extreme cases, maybe even lead to extinction. The effect would be strongest for species exhibiting important, transient reductions in habitability, such as Thunnus atlanticus after Fig. 5b, and for species exhibiting low migration potentials. I would encourage the authors to briefly discuss this potential limitation in section 4 (discussion and conclusions).

**4.** Species limitation towards the cold edge. My understanding is that the aerobic growth index suffers of the same limitations as the metabolic index with regards to its incapacity to mechanistically limit the extension of the species in the high latitudes, where [O2] is high and temperatures low. This would bias the distribution of the species (and the latitudinal diversity gradient). Is that the case? Was instead some empirical limitation added like Deutsch et al. (2022; https://doi.org/10.1126/science.abe9039) did? I think this limitation should be shortly discussed, just for the reader to be aware of this aspect.

**Minor and technical comments**:

- Lines 73–74: what is "lithogenic material" referring to, here?
- Lines 138 and 139: Here we read 296 years, while 269 appears elsewhere (e.g., caption of Figs. 2, 5, A2), please check. Also, I did not understand how this duration was established, would it be possible to expand on / clarify this point on lines 136–140?
- Lines 183–184: is there any technical reason for not just using the 3D-dimensional model output?
- Line 210: This section should be called "Results"
- Line 215: would it be possible to also provide rough equivalents of $pO_2$ values (in mbar) in terms of oceanic [$O_2$] (in micro mol / kg or similar), for reader more familiar with these units?
- Line 225: reference to panels (g,h) is missing.
- Line 228: please revise ("species' their").
- Line 272: "drivers"
- Caption of Fig. 2: since AGIrel is a difference as such, wouldn't it be more rigorous to write "as the difference AGIrel between…"
- Line 295: "increased ventilation and reduced biological oxygen consumption". This combination, although well demonstrated based on the figures, is counter-intuitive. Is it because increased convection induces the advection of low-nutrient waters coming from the low latitudes, similar to Rae et al. (2020; https://www.science.org/doi/10.1126/sciadv.abd1654)?
- Line 406: what does "reversibility timescales" refer to?
- Lines 414–415: "our findings align with variable species responses seen in distribution shifts". If this statement is supported, I think it would be a great addition to expand a bit on this. Otherwise, I would suggest deleting this sentence which, without additional details, is not very convincing.

---

## Referee Comment (RC2)

**Review report**

**Long-term impacts of global temperature stabilization and overshoot on exploited marine species**

Anne L. Morée, Fabrice Lacroix, William W. L. Cheung and Thomas L. Frölicher

**Summary of the manuscript**

The manuscript uses the GFDL-ESM2M Earth System Model to examine the impact of climate warming scenarios on marine ecosystems based on the Aerobic Growth Index (AGI). They analyze how marine ecosystem habitats are affected under the overshoot vs. stabilization scenarios and quantifies how organisms' adaptation capabilities may mitigate habitat loss due to climate change. They find that under the overshoot scenario, although temperatures peak at 2°C and stabilize at 1.5°C, the maximum habitat loss still occurs over 150 years after the warming peak. Furthermore, they demonstrates that organisms' adaptation strategies will have a measurable impact on capping habitat loss under warming, depending on the adaptation rate.

**Major review**

The manuscript is generally well-written and structured. The authors outline a clear research question and provide nearly sufficient evidence to support it. I particularly appreciate the attempt to predict adaptability within this approach; although not extensive, it will be a valuable contribution. Adaptation has typically not been quantitatively applied in these types of studies.

That said, there are a couple of limitations inherent to the study's approach that warrant clarification. First, the authors chose to use the relative AGI formulation, which, by definition, removes the species-scale oxygen supply versus demand. Because of the elimination of physiological oxygen dependence, the relative AGI only provides a general "sense" of ecosystem sensitivity to a warming climate, rather than an objective quantitative prediction. While this formulation is mathematically useful and allows for a more flexible analytical framework, it is unclear whether the relative AGI remains instructive for quantifying changes in species abundance. The argument here is that habitat gain or loss with changing environmental conditions depends on organisms' metabolic resource management (the ability to sustain respiration), which varies among species. Thus, vulnerability to stress is also variable among species, which is a key aspect of ecosystem resilience. Namely, not all species are equally important or vulnerable; the loss of some species is more detrimental to the ecosystem than others. Therefore, to appropriately quantify long-term changes in species abundance, it is necessary to assess species' vulnerability to environmental stress. To my understanding, species-vulnerability information is excluded when using the relative AGI, habitability is estimated by a generic count. Thus, it is not clear that the "sense" of habitat changes with warming based on relative AGI is quantitatively meaningful beyond serving as an indicator for potentially hazardous and habitable regions under a warming climate.

For example, during environmental stress events like heat waves or cold spells, it is the extremes of the physiological trait space that matter most for the entire system's ecosystem response and survival. Although the authors mention some of these limitations, they remain quantitative. My main point can be summarized as follows: while relative AGI serves as a meaningful metric for understanding climate change's impact on marine ecosystems by integrating oxygen and temperature changes into a single ecosystem-relevant proxy, it may not be sufficient quantitatively as an indicator of changes in ecosystem abundance. Perhaps I'm missing something; please clarify.

Secondly, the authors repeatedly mention the advantages of AGI over the classical metabolic index proposed by Deutsch et al., citing differences in requirements for lab-based physiological data. However, this difference appears to stem from the application of relative AGI, which removes physiological constraints. I believe the metabolic index can also be applied in a similar manner by simplifying it to yield a relative metabolic index that eliminates physiological parameters. Thus, it is not clear how physiological requirement data differ between classical AGI and the metabolic index; perhaps I am missing something here again; please clarify.

Beyond these two points, the manuscript is well-written and would be an important contribution to the community.

**Below are some minor and specific comments**

This is a minor point, while I appreciate the application of adaptability in marine organisms, it would be helpful to briefly discuss how robust this adaptation methodology is (or discuss the limitations). While this time-based adaptation may as well reflect how marine species will develop resilience against warming stress, ecosystems often experience tipping points leading to significant habitat loss or gain; such changes can tip systems on short timescale when important species are lost or gained. Moreover, tipping point climate variability and extremes occur over much shorter timescales. I'm curious to how you think about this, it might be worth clarifying.

**Line 25**: Missing. Howard et al., 2019; Mongwe et al., 2024.

**Line 50**: This description is unclear and seems incomplete. Perhaps first define what AGI is or move this section to the methods where you can define things more clearly. At this stage, the text assumes the reader knows what AGI is, which is not a reasonable expectation. You only define AGI later in the text.

**Line 165**: Here you say that relative AGI gives a "sense of the direction and magnitude of habitability." Maybe only the direction, but not the magnitude; it is not still clear to me that you can quantify change without information on organisms' physiological constraints as stated above. Maybe you can, please clarify.

**Line 170**: Once again, "AGI can be a poor indicator of which specific species thresholds are needed." How do you make this distinction when this information is needed or not to make a good prediction? When you don't include this information, how do you know the magnitudes of change are robust?

**Line 219**: This description is not clear; the text requires familiarity with Logan et al., 2014. I had to refer to Logan et al., 2014 to understand your point here. Perhaps rephrase and make it a bit clearer

**References**

Howard, E. M., Penn, J. L., Frenzel, H., Seibel, B. A., Bianchi, D., Renault, L., Kessouri, F., Sutula, M. A., Mcwilliams, J. C., and Deutsch, C.: Climate-driven aerobic habitat loss in the California Current System, Sci. Adv., 6, eaay3188, https://doi.org/10.1126/sciadv.aay3188, 2020.

Mongwe, P., Long, M., Ito, T., Deutsch, C., and Santana-Falcón, Y.: Climatic controls on metabolic constraints in the ocean, Biogeosciences, 21, 3477–3490, https://doi.org/10.5194/bg-21-3477-2024, 2024.

---

## Author Comment (AC1)

**Response to the editor and reviewers**

We thank the editor and the two reviewers for the critical assessment of our work and their very helpful and constructive comments. We have addressed all comments point by point and will revise our manuscript accordingly.

**Reviewer 1**

**Paper summary**:

Morée et al. quantify the impact of ocean warming and deoxygenation on the habitability of the ocean by 46 exploited marine species until year 2500. The authors consider temperature stabilization scenarios and overshoot scenarios, with temperature in the latter case reaching a maximum before decreasing and stabilizing. Ocean habitability is calculated using the aerobic growth index and environmental conditions simulated in transient simulations conducted with the GFDL ocean- atmosphere general circulation model. Results demonstrate that only around half of the total habitat loss is realized when target warming levels are reached. Habitat loss continues in the decades to centuries after that. Species adaptation may lower the overall habitat loss.

**General comment**:

The manuscript represents a welcome departure over previous work. It is clearly written and illustrated with figures of excellent quality that efficiently convey key information supporting the text. The methods require lots of attention and are not always easy to read, but this is for the better since the setup is described in detail and I do not think this aspect can be improved without being detrimental to the scientific content of the manuscript. The results are also very well organized, starting with describing the overall changes in aerobic growth index, then the underlying environmental / climatic drivers, before focusing on changes in habitability at the species level and finishing with an analysis of the impact of adaptation, overall constituting a very informative and comprehensive work. I encourage the publication after very minor revisions and provide suggestions of improvements below.

We thank the reviewer for the positive assessment and detailed comments, which we will incorporate to further improve our manuscript.

Please note that I was not able to try and run the code, which was not provided with the manuscript (with a simple placeholder found in the 'code and data availability' section).

The code underlying the analysis will be uploaded onto a Zenodo repository and the link added to the code and data availability section.

**Main comments**:

1. Selection of the 46 species. It would be beneficial to provide a short description of the species selected in the Methods: what are these species, and how were they selected? A reader being not familiar with previous work cited would better understand the context of the study.

We will add a link to Figure 5 to the revised manuscript, where the species names are listed. The selection of species was partly described in section 2.6, but this description will be extended to: *"Spatial distribution data for the 46 representative exploited species (species names are indicated in Figure 5; Morée et al., 2023; Palomares et al., 2004) are used to calculate $T_{pref,i}$, $pO_2$,threshold, and $AGI_{crit}$. These data also form the reference habitat for assessing changes in contemporary habitat volume. The species were selected such that they provide a representative range in body size, climatic zone (tropical, temperate), habitat size, and depth range. The 46 species also cover a broad range of vulnerabilities to warming and deoxygenation, with the most vulnerable species having a ~30 times larger change in volume per unit change in AGI than the least vulnerable species (Morée et al., 2023). We include 23 species with their predominant occurrence in the epipelagic (0-200m depth), 5 species that mostly inhabit the mesopelagic (200-1000m depth) as well as 18 demersal species which live on or just above the sea floor (for which we use the deepest ocean model layer). Some pelagic and deep-water wide-ranging species were selected that inhabit both tropical and temperate regions."*

2. Selection of the climatic scenarios. Similarly, it may be instructive to tell a bit more about the choice of the stabilization scenarios. How were the magnitude and time scales of temperature change selected? It may be good to discuss these numbers with regards to the IPCC scenarios. Regarding the overshoot scenarios, it may be interesting to discuss the plausibility of the time scale of overshoot and subsequent stabilization: is the time scale plausible, based on previous work that did not force temperature changes but mechanistically simulated them, e.g. using sediment-enabled earth system models of intermediate complexity? Such discussion would better frame the context of the work and hence strengthen the study.

We thank the reviewer for this important point. We will add to the method section 2.5: *"The temperature changes were selected to align with the global warming levels commonly used in the IPCC reports - 1.5 °C, 2.0 °C and 3.0 °C (IPCC, 2022) - as benchmarks for assessing impacts. Additionally, we included a stabilization scenario at the current level of warming to investigate committed impacts. The warming rates leading to these temperature levels closely follow (at least initially) those observed during the historical period."*

We will add a paragraph in the discussion section on the length of overshoot scenarios, and their plausibility:

*"Commonly used CMIP6 overshoot scenarios projected a rapid rise to around 2°C of global warming, followed by reversal within 20-50 years (Pfleiderer et al., 2024). However, such a rapid reversal, requiring substantial atmospheric $CO_2$ removal, is unlikely given current limitations in carbon dioxide removal technologies (Schleussner et al., 2024). In our 3°C overshoot scenario, where we assume that $CO_2$ removal occurs over more than 150 to 200 years, negative $CO_2$-fe emissions of up to 9 Pg C $yr^{-1}$ would still be required to bring temperatures back to 1.5°C (Fig. A1). This scale of removal far exceeds what is achievable in the foreseeable future (Fuss et al., 2018). Even in the 2°C overshoot scenario, the peak negative $CO_2$ emissions would need to reach approximately 3 Pg C $yr^{-1}$."*

3. Infinite migration potential. An implicit assumption of the approach is that studied species can always populate the regions of the ocean where oxygen supply is sufficient for them. In reality, we might imagine that some species would not be able to keep up with changing environmental

conditions due to limited migration potentials, which may impact the calculated habitability and reversibility and, in some extreme cases, maybe even lead to extinction. The effect would be strongest for species exhibiting important, transient reductions in habitability, such as Thunnus atlanticus after Fig. 5b, and for species exhibiting low migration potentials. I would encourage the authors to briefly discuss this potential limitation in section 4 (discussion and conclusions).

Thank you for the suggestions. We agree with the review to clarify and discuss this assumption in the approach. We will add the following to the method and discussion sections of the revised manuscript.

In the method section 2.5: "*We recognize the potential habitat volume and the reversibility of their changes may not be realized by the species because of other biogeographic constraints such as dispersal potentials, availability of suitable prey or other environmental limitations beyond temperature and oxygen.*"

In the discussion section 4: "*Moreover, the impacts of changing temperature and oxygen levels on species' habitat volume may be underestimated under the overshoot scenarios. Specifically, species may not be able to shift to viable habitats because of biogeographic constraints not represented by the AGI e.g., dispersal potential, trophic interactions. Such uncertainties are particularly notable for species with large transient changes in habitat volume such as blackfin tuna (Thunnus atlanticus) and those with relatively lower dispersal potential e.g., coral hind (Cephalopholis miniata).*"

4. Species limitation towards the cold edge. My understanding is that the aerobic growth index suffers of the same limitations as the metabolic index with regards to its incapacity to mechanistically limit the extension of the species in the high latitudes, where [O2] is high and temperatures low. This would bias the distribution of the species (and the latitudinal diversity gradient). Is that the case? Was instead some empirical limitation added like Deutsch et al. (2022; https://doi.org/10.1126/science.abe9039) did? I think this limitation should be shortly discussed, just for the reader to be aware of this aspect.

Thank you for the suggestion. We agree that the version of AGI used in this manuscript does not represent species limitation towards the cold edge. This will now be discussed in the revised manuscript in the discussion section: "*Furthermore, the AGI, in its current form, only represents the limitation of temperature and oxygen on the warm-temperature edge of fish distributions and does not represent the observed reduction in aerobic scope at the low temperature edge (Pörtner 2010; Clarke et al. 2021; Deutsch et al. 2022). Thus, our results do not include habitat expansion or contraction due to ocean warming or cooling at the cold edge of species distributions. Development and application of AGI that incorporate low temperature limitation of species' metabolism and growth will help quantify such uncertainties.*"

**Minor and technical comments**:

Lines 73–74: what is "lithogenic material" referring to, here?

We refer here to particulate matter. The TOPAZv2 module of the GFDL ESM2M simulates the full cycle of particulate matter inputs, dynamics and sediment deposition and burial. We will clarify this in the manuscript.

Lines 138 and 139: Here we read 296 years, while 269 appears elsewhere (e.g., caption of Figs. 2, 5, A2), please check. Also, I did not understand how this duration was established, would it be possible to expand on / clarify this point on lines 136–140?

*Thank you for spotting this. It should be 296 years and will be corrected throughout the MS. We will also clarify: "Since the 3°C stabilization simulation reaches its warming level the latest – specifically in the year 2204 – there are 296 years remaining until the end of the simulation in 2500. Consequently, we adopt this 296-year timescale for assessing committed changes in all stabilization scenarios."*

Lines 183–184: is there any technical reason for not just using the 3D-dimensional model output?

*We will clarify with the following: "We acknowledge that some species may occupy only part of their assigned depth range or may temporarily reside outside it, either above or below. Nevertheless, we believe that the assigned depth ranges generally provide a reasonable estimate of in-habitat $pO_2$ and temperature variability. The reason for not just using the 3D model output across the entire depth range of each species' depth realm is primarily due to the current lack of reliable 3D species distributions for our selected species."*

Line 210: This section should be called "Results"

*This will be changed.*

Line 215: would it be possible to also provide rough equivalents of $pO_2$ values (in mbar) in terms of oceanic [$O_2$] (in micro mol / kg or similar), for reader more familiar with these units?

*We think it might be confusing to suddenly come up with different units only in one location in the manuscript. We also note that the AGI depends on the partial pressure of $O_2$ and not on the concentration of $O_2$. Therefore, we keep the text as is.*

Line 225: reference to panels (g,h) is missing.

*This will be added.*

Line 228: please revise ("species' their").

*The sentence will read: "Thick lines show the median across the individual species' 31-year running mean data and thin lines annual mean time series."*

Line 272: "drivers"

*Done.*

Caption of Fig. 2: since AGIrel is a difference as such, wouldn't it be more rigorous to write "as the difference AGIrel between..."

*We show the difference in AGI$^{rel}$ between different time periods. Therefore, we will keep the sentences as written.*

Line 295: "increased ventilation and reduced biological oxygen consumption". This combination, although well demonstrated based on the figures, is counter-intuitive. Is it because increased convection induces the advection of low-nutrient waters coming from the low latitudes, similar to Rae et al. (2020; https://www.science.org/doi/10.1126/sciadv.abd1654)?

We agree with the reviewer that this result appears counterintuitive. However, investigating the reasons behind the decrease in biological oxygen consumption and whether it is linked to the advection of low-nutrient waters from lower latitudes lies beyond the scope of this paper. Nonetheless, we will consider this valuable insight for potential follow-up analysis.

Line 406: what does "reversibility timescales" refer to?

We will revise the sentence as follows: ".. since changes in ocean conditions in greater depths take substantially longer to reverse following overshoot scenarios (Santana-Falcon et al. 2023; Schwinger et al. 2022)."

Lines 414–415: "our findings align with variable species responses seen in distribution shifts". If this statement is supported, I think it would be a great addition to expand a bit on this. Otherwise, I would suggest deleting this sentence which, without additional details, is not very convincing.

We will delete it.

References:

Fuss, S., Lamb, W. F., Callaghan, M. W., Hilaire, J., Creutzig, F. *et al.* (2018). Negative emissions - Part 2: Costs, potential and side effects. *Environmental Research Letters* , 13, 063002. https://doi.org/10.1088/1748-9326/aabf9f

IPCC, 2022: Summary for Policymakers [H.-O.Pörtner, D.C.Roberts, E.S.Poloczanska, K.Mintenbeck, M.Tignor, A. Alegría, M. Craig, S. Langsdorf, S. Löschke, V. Möller, A. Okem (eds.)]. In: *Climate Change 2022: Impacts, Adaptation and Vulnerability.* Contribution of Working Group II to the Sixth Assessment Report of the Intergovernmental Panel on Climate Change [H.-O.Pörtner, D.C.Roberts, M.Tignor, E.S.Poloczanska, K.Mintenbeck, A.Alegría, M.Craig, S. Langsdorf, S. Löschke, V. Möller, A. Okem, B. Rama (eds.)]. Cambridge University Press, Cambridge, UK and New York, NY, USA, pp. 3–33, doi:10.1017/9781009325844.001.

Pfleiderer, P., Schleussner, C.-F., Sillmann, J. (2018). Limited reversal of regional climate signals in overshoot scenarios. *Environmental Research Letters*, 3, 015005. https://doi.org/10.1088/2752-5295/ad1c45

Schleussner, CF., Ganti, G., Lejeune, Q. *et al.* (2024). Overconfidence in climate overshoot. *Nature*, 634, 366–373. https://doi.org/10.1038/s41586-024-08020-9

---

## Author Comment (AC2)

**Response to the editor and reviewers**

We thank the editor and the two reviewers for the critical assessment of our work and their very helpful and constructive comments. We have addressed all comments point by point and will revise our manuscript accordingly.

**Reviewer 2**

**Summary of the manuscript**

The manuscript uses the GFDL-ESM2M Earth System Model to examine the impact of climate warming scenarios on marine ecosystems based on the Aerobic Growth Index (AGI). They analyze how marine ecosystem habitats are affected under the overshoot vs. stabilization scenarios and quantifies how organisms' adaptation capabilities may mitigate habitat loss due to climate change. They find that under the overshoot scenario, although temperatures peak at 2°C and stabilize at 1.5°C, the maximum habitat loss still occurs over 150 years after the warming peak. Furthermore, they demonstrates that organisms' adaptation strategies will have a measurable impact on capping habitat loss under warming, depending on the adaptation rate.

**Major review**

The manuscript is generally well-written and structured. The authors outline a clear research question and provide nearly sufficient evidence to support it. I particularly appreciate the attempt to predict adaptability within this approach; although not extensive, it will be a valuable contribution. Adaptation has typically not been quantitatively applied in these types of studies.

We thank the reviewer for the positive assessment and detailed comments, which further improved our manuscript.

That said, there are a couple of limitations inherent to the study's approach that warrant clarification. First, the authors chose to use the relative AGI formulation, which, by definition, removes the species-scale oxygen supply versus demand. Because of the elimination of physiological oxygen dependence, the relative AGI only provides a general "sense" of ecosystem sensitivity to a warming climate, rather than an objective quantitative prediction. While this formulation is mathematically useful and allows for a more flexible analytical framework, it is unclear whether the relative AGI remains instructive for quantifying changes in species abundance. The argument here is that habitat gain or loss with changing environmental conditions depends on organisms' metabolic resource management (the ability to sustain respiration), which varies among species. Thus, vulnerability to stress is also variable among species, which is a key aspect of ecosystem resilience. Namely, not all species are equally important or vulnerable; the loss of some species is more detrimental to the ecosystem than others. Therefore, to appropriately quantify long-term changes in species abundance, it is necessary to assess species' vulnerability to environmental stress. To my understanding, species-vulnerability information is excluded when using the relative AGI, habitability is estimated by a generic count. Thus, it is not clear that the "sense" of habitat changes with warming based on relative AGI is quantitatively meaningful beyond serving as an indicator for potentially hazardous and habitable regions under a warming climate.

For example, during environmental stress events like heat waves or cold spells, it is the extremes of the physiological trait space that matter most for the entire system's ecosystem response and survival. Although the authors mention some of these limitations, they remain quantitative. My main point can be summarized as follows: while relative AGI serves as a meaningful metric for understanding climate change's impact on marine ecosystems by integrating oxygen and temperature changes into a single ecosystem-relevant proxy, it may not be sufficient quantitatively as an indicator of changes in ecosystem abundance. Perhaps I'm missing something; please clarify.

*Thank you for the comment. In the manuscript, we applied the relative AGI that does not represent species-specific biological information to indicate ecosystem-level vulnerabilities to the impacts of ocean warming and deoxygenation. However, we also calculated species-specific AGI (see first paragraph of section 2.5 and equation 1) to examine the potential impacts on viable habitat volume at the species level (see Fig. 5, A3 and A4). In the latter application, species-specific oxygen thresholds were calculated and used from biogeographical data. The entire section 3.3 'Impacts on contemporary habitat volume of individual species' describes and discusses the species-specific results.*

*To further clarify the different formulations and application of AGI, we will clarify this at the beginning of section 2.5: "We applied two alternative formulations of AGI as indicators of species-level and ecosystem level vulnerabilities to the impacts of ocean warming and deoxygenation (see eq. 1 and 2, respectively)."*

Secondly, the authors repeatedly mention the advantages of AGI over the classical metabolic index proposed by Deutsch et al., citing differences in requirements for lab-based physiological data. However, this difference appears to stem from the application of relative AGI, which removes physiological constraints. I believe the metabolic index can also be applied in a similar manner by simplifying it to yield a relative metabolic index that eliminates physiological parameters. Thus, it is not clear how physiological requirement data differ between classical AGI and the metabolic index; perhaps I am missing something here again; please clarify.

*As discussed above, we used species specific thresholds. Also, we think that the comparison with metabolic index is not directly relevant in the discussion. Thus, we will remove the sentences in the revised manuscript to avoid confusion.*

*"Relative changes in $pO_2$ supply over $pO_2$ demand ratios have been implied to assess ecosystem-level impacts (Battaglia & Joos, 2018; Deutsch, Ferrel, Seibel, et al., 2015; Oschlies, 2021; Santana-Falcón et al., 2023), but species-specific thresholds and preference windows are needed for such estimates (Morée et al., 2023).  In the future, the application of the AGI to more species will allow the assessment of a wider range of interspecies responses."*

Beyond these two points, the manuscript is well-written and would be an important contribution to the community.

**Below are some minor and specific comments**

This is a minor point, while I appreciate the application of adaptability in marine organisms, it would be helpful to briefly discuss how robust this adaptation methodology is (or discuss the limitations). While this time-based adaptation may as well reflect how marine species will develop resilience against warming stress, ecosystems often experience tipping points leading to significant habitat loss or gain; such changes can tip systems on short timescale when important species are lost or gained. Moreover, tipping point climate variability and extremes occur over much shorter timescales. I'm curious to how you think about this, it might be worth clarifying.

Thank you for highlighting this important research question about the role of tipping points/events in species adaptation. This is one of many critical questions regarding how adaptation shapes biodiversity and biogeography under global change. For instance, some studies suggest that rapid environmental changes impose strong selection pressures on organisms, potentially driving adaptation or extinction (e.g., Grant et al. 2017). However, due to the scarcity of data on tipping events and evolutionary responses, these dynamics remain challenging to study.

Our analysis represents a first step in exploring the potential implications of adaptation for marine biogeography under centennial-scale climate change. We hope this paper inspires future research to delve deeper into these complex and pressing issues.

To clarify this, we will add to the discussion in the revised manuscript: "*In addition, one of many critical questions regarding how adaptation shapes biodiversity and biogeography under global change is the role of extreme events and tipping points in species adaptation. However, these dynamics remain challenging to study due to the scarcity of data on tipping events and evolutionary responses (Grant et al. 2017). Our analysis represents a first step in exploring the potential implications of adaptation for marine biogeography under centennial-scale climate change. Our findings could inspire future research to delve deeper into these complex and pressing issues.*"

**Line 25**: Missing. Howard et al., 2019; Mongwe et al., 2024.

We thank the reviewer for pointing us to those two important references, which will be included in the revised text.

**Line 50**: This description is unclear and seems incomplete. Perhaps first define what AGI is or move this section to the methods where you can define things more clearly. At this stage, the text assumes the reader knows what AGI is, which is not a reasonable expectation. You only define AGI later in the text.

We will define AGI when the term is first mentioned in the manuscript.

"*The AGI indicates the potential habitat conditions that theoretically support the aerobic scope required for the growth of marine water-breathing ectotherms, represented by the ratio of $pO_2$ supply to metabolic demand.*"

**Line 165**: Here you say that relative AGI gives a "sense of the direction and magnitude of habitability." Maybe only the direction, but not the magnitude; it is not still clear to me that you can

quantify change without information on organisms' physiological constraints as stated above. Maybe you can, please clarify.

As explained in the response above, in the manuscript, we did calculate species-specific AGI. Critical partial oxygen threshold for each species was calculated from biogeographical data as described in Clarke et al. (2021).

**Line 170**: Once again, "AGI can be a poor indicator of which specific species thresholds are needed." How do you make this distinction when this information is needed or not to make a good prediction? When you don't include this information, how do you know the magnitudes of change are robust?

This sentence refers to the fact that the relative AGI does not account for species-specific thresholds. To avoid confusion, we will delete this sentence.

**Line 219**: This description is not clear; the text requires familiarity with Logan et al., 2014. I had to refer to Logan et al., 2014 to understand your point here. Perhaps rephrase and make it a bit clearer

We will clarify this in the manuscript by noting that the approach used by Logan et al. (2014) to represent adaptation is described in detail in the subsequent paragraph: "*We thereby follow the approach applied to corals by Logan et al. (2014), as outlined in the paragraphs below.*"

Reference:
Grant, P. R., Grant, B. R., Huey, R. B., Johnson, M. T. J., Knoll, A. H., and Schmitt, J. (2017). Evolution caused by extreme events. *Philosophical Transactions of the Royal Society B: Biological Sciences*, 372(1723), 20160146. https://doi.org/10.1098/rstb.2016.0146